# Deep Transcriptome Analysis Reveals Reactive Oxygen Species (ROS) Network Evolution, Response to Abiotic Stress, and Regulation of Fiber Development in Cotton

**DOI:** 10.3390/ijms20081863

**Published:** 2019-04-15

**Authors:** Yanchao Xu, Richard Odongo Magwanga, Xiaoyan Cai, Zhongli Zhou, Xingxing Wang, Yuhong Wang, Zhenmei Zhang, Dingsha Jin, Xinlei Guo, Yangyang Wei, Zhenqing Li, Kunbo Wang, Fang Liu

**Affiliations:** 1State Key Laboratory of Cotton Biology, Institute of Cotton Research, Chinese Academy of Agricultural Sciences (ICR, CAAS), Anyang 455000, China; xuyanchao2016@163.com (Y.X.); magwangarichard@yahoo.com (R.O.M.); caixy@cricaas.com.cn (X.C.); zhouzl@cricaas.com.cn (Z.Z.); wangxx@cricaas.com.cn (X.W.); wangyh2525377@163.com (Y.W.); xyxuezhihua@163.com (Z.Z.); jindingsha@163.com (D.J.); guoxlcaas@163.com (X.G.); weiyangyang511@126.com (Y.W.); lizhenqing2019@163.com (Z.L.); 2Jaramogi Oginga Odinga University of Science and Technology (JOOUST), School of Biological and Physical Sciences (SPBS), P.O BOX 210-40600, Bondo 210-40600, Kenya; 3Biological and Food Engineering, Anyang Institute of Technology, Anyang 455000, China

**Keywords:** *ROS* genes, transcription analysis, abiotic stress, fiber development

## Abstract

Reactive oxygen species (ROS) are important molecules in the plant, which are involved in many biological processes, including fiber development and adaptation to abiotic stress in cotton. We carried out transcription analysis to determine the evolution of the *ROS* genes and analyzed their expression levels in various tissues of cotton plant under abiotic stress conditions. There were 515, 260, and 261 genes of ROS network that were identified in *Gossypium hirsutum* (AD_1_ genome), *G. arboreum* (A genome), and *G. raimondii* (D genome), respectively. The *ROS* network genes were found to be distributed in all the cotton chromosomes, but with a tendency of aggregating on either the lower or upper arms of the chromosomes. Moreover, all the cotton *ROS* network genes were grouped into 17 families as per the phylogenetic tress analysis. A total of 243 gene pairs were orthologous in *G. arboreum* and *G. raimondii*. There were 240 gene pairs that were orthologous in *G. arboreum*, *G. raimondii*, and *G. hirsutum*. The synonymous substitution value (*K*_s_) peaks of orthologous gene pairs between the At subgenome and the A progenitor genome (*G. arboreum*), D subgenome and D progenitor genome (*G. raimondii*) were 0.004 and 0.015, respectively. The *K*_s_ peaks of ROS network orthologous gene pairs between the two progenitor genomes (A and D genomes) and two subgenomes (At and Dt subgenome) were 0.045. The majority of *K*_a_/*K*_s_ value of orthologous gene pairs between the A, D genomes and two subgenomes of TM-1 were lower than 1.0. RNA seq. analysis and RT-qPCR validation, showed that, CSD1,2,3,5,6; FSD1,2; MSD1,2; APX3,11; FRO5.6; and RBOH6 played a major role in fiber development while CSD1, APX1, APX2, MDAR1, GPX4-6-7, FER2, RBOH6, RBOH11, and FRO5 were integral for enhancing salt stress in cotton. ROS network-mediated signal pathway enhances the mechanism of fiber development and regulation of abiotic stress in Gossypium. This study will enhance the understanding of ROS network and form the basic foundation in exploring the mechanism of ROS network-involving the fiber development and regulation of abiotic stress in cotton.

## 1. Introduction

*Gossypium* is one of the largest and most widely distributed genus with more than 50 species [1,2]. Cotton has a long evolution and domestication history. *Gossypium* diverged from its relatives approximately 10–15 million years ago (MYA), and evolved into three major diploid lineages, the New World clade, the African-Asian clade and the Australian clade, about 5–10 MYA [3]. The allopolyploid cottons emerged about 1–2 MYA due to an intergenomic hybridization event between A and D genomes [3,4], and were domesticated at least 4000 to 5000 years ago [5]. Both *G. arboreum* and *G. hirsutum* have a natural long spinable fiber, although, *G. raimondii* also generates fiber, its fibers are short and not spinable [6]. During human domestication history of cotton, the central focus was on the fiber quality and quantity [7]. The changes in the environment has resulted in to paradigm shift not only to explore on fiber and fiber attributes, but also on the survival strategies developed by plants, whereas plants try to adapt to various environmental conditions in the course of their evolutionary history. Thus, positive selections of genes were different during evolution and human domestication. Increasing number of genome sequencing and resequencing, mRNA sequencing and phenotype assessment of cotton [8] provided an important resource for studying potential biological mechanism related to abiotic stress tolerance in cotton. 

Reactive oxygen species (ROS), including singlet oxygen (^1^O_2_), superoxide anion (O_2_^−^), hydrogen peroxide (H_2_O_2_), and hydroxyl radical (•OH), produced by many of the metabolic pathways in aerobic cells, most likely appeared on Earth together with the first atmospheric oxygen molecules about 2.4–3.8 billion years ago and have been a constant companion of aerobic life [9]. Many biological processes are controlled by ROS, which include but are not limited to programmed cell death, biotic and abiotic stress responses, and plant growth and development [10]. However, ROS have been thought to be toxic and unwanted molecules for a long time, but now are widely known as indispensable molecule for plant lives [11]. ROS level is regulated by an intricate network composed of at least 152 genes in Arabidopsi*s* [12,13,14,15], but ROS regulation, mechanisms and the genes involved is not clear in cotton. The ROS regulation network consists of genes which functions as inductors of enzymes which do scavenge on the ROS molecules thereby reducing their toxicity to the plant cell [12]. 

Cotton fiber development and abiotic stress responses are controlled by many molecules and complex pathways, including ROS [16]. The evolution of long spinnable fibers in cotton is accompanied by novel expression of genes assisting in the regulation of ROS levels [17]. A calcium sensor, *GhCaM7*, might modulate ROS production and act as a molecular link between Ca^2+^ and ROS signal pathways in early fiber development stages [18]. ROS-producing genes, including gene encoding NADPH oxidases/respiratory burst oxidase homologue (RBOH), regulated cell expansion through activation of Ca^2+^ channels, and are also involved in ROS regulation in the plant cell [19,20]. The detection of these genes found to have an integral role in the ROS regulation and fiber development, clearly shows that ROS-scavenging pathway may also be important in the regulation of fiber development. A cytosolic ascorbate peroxidase (APX) gene, *GhAPX1AT/DT*, regulating optimal H_2_O_2_ level, is the key mechanism regulating fiber elongation and deregulation of the increase in H_2_O_2_ results in shorter fiber by initiating secondary cell wall-related gene expression [21]. An ascorbate oxidase (AO) gene, *GhAO1*, controls cotton fiber elongation via the auxin-mediated signaling pathway [22]. A NAC transcription factor, *GhFSN1*, acts as a positive regulator in controlling second cell wall formation of cotton fiber [23]. Many of the abiotic stresses cause oxidative stress and genes of ROS network also involve in abiotic stress responses [24]. Furthermore, transcriptome analysis of an upland cotton, *G. hirsutum*, has provided the basis to elucidate the various mechanisms adopted by plants in dealing with salt stress [25].

In this study, genes in ROS network were identified in cotton. Evolutionary analysis proved that ROS network was a conservative and stable system during *Gossypium* evolution and domestication. Genes of the ROS network that are involved in fiber development and abiotic stress were detected in this research work. A simple ROS signal pathway was used to describe the potential mechanism of fiber development and regulation of abiotic stress in *Gossypium*. In summary, this study will enhance the understanding of ROS network and pave the path in studying the mechanism of ROS network-involving fiber development and regulation of abiotic stress in cotton.

## 2. Results

### 2.1. ROS Network Genes Identification in Gossypium Species

A total of 515, 260, and 261 ROS candidate proteins were identified in *G. hirsutum*, *G. arboreum*, and *G. raimondii*, respectively (Appendix A). The proportions of the ROS proteins obtained in the three cotton species, supports the evolution theory of the tetraploid cotton. The allotetraploid cotton, *G. hirsutum* harbored twice the number individual numbers of the ROS proteins obtained for either of the two diploid cotton species. In all the ROS proteins obtained from the three cotton species, they were further subdivided into different classes or sub families, in which the highest ROS proteins were observed for peroxidase (PER) with 279 proteins encoding the ROS genes, accounting for 26.9% of all the ROS proteins in the three cotton species. These proteins included 36 superoxide dismutase (SOD), 47 ascorbate peroxidase (APX), 24 monodehydroascorbate reductase (MDAR), 20 dehydroascorbate reductase (DHAR), 12 glutathione reductase (GR), 17 catalase (CAT), 32 glutathione peroxidase (GPX), 21 ferritin (FER), 58 respiratory burst oxidase homologue (RBOH), 27 NADPH-like oxidase (FRO), 19 Ubiquinol oxidase (AOX), 37 peroxiredoxin (PrxR), 224 thioredoxin reductase (Trx), 183 Glutaredoxin (GLR) and 279 peroxidase (PER). Beside NADPH oxidases and NADPH oxidase-like (ROS producer), all of the genes encode ROS-scavenging enzymes. There are 152 genes, so far identified in Arabidopsis. Moreover, in genome-wide identification of the *ROS* genes in Mizuna plants, a total of 32 *ROS* genes were identified out of the 22,428 transcript, in which 8258 and 14,170 transcripts were up- and down-regulated, respectively [26]. The high number of the *ROS* genes in *Gossypium*, suggested a more complex ROS network in relation to fiber development and enhancing abiotic stress tolerance. 

### 2.2. Chromosome Mapping of the ROS Network Gene Family

The *ROS* genes were mapped and found to be widely distributed across the entire cotton chromosomes. In tetraploid cotton, *G. hirsutum*, the *ROS* genes were mapped in all the 26 chromosomes, in which the highest gene loci density were observed in chromosome At05 and its homeolog chromosome with 46 (18.11%) and 38 (14.28%) *ROS* genes, respectively, while the lowest gene loci density in chromosome At04 with only 10 *ROS* genes accounting for 1.94% of all the *ROS* genes identified in *G. hirsutum* (Figure 1A). The gene distribution among the diploid cotton species, *G. arboreum* and *G. raimondii* were found to be present in each of the 13 chromosomes, in which the highest gene loci were detected in chromosome A10 (35, 12.36%) and D09 (40, 13.89%) in *G. arboreum* and *G. raimondii*, respectively (Figure 1B,C). Moreover, the genes mapping in the individual chromosomes showed the tendencies of the genes aggregating on either upper or the lower arms of the chromosomes, similar observation have been previously observed in the distribution of the *LEA* genes in cotton, with high gene loci densities being observed in chromosomes A10 and D09 [27]. The detection of the high number of the *ROS* genes within the chromosomes showed that the three sets of chromosomes harbor significant genes with diverse roles in enhancing abiotic stress tolerance in plants. This result suggests that chromosome D09, chromosome Dt05 and its homeolog chromosome At05 play a core role in the evolution and dynamic balance in the *ROS* genes balance in cotton.

### 2.3. Phylogenetic Tree, Gene Duplication, and Synteny Analysis of the ROS Network Gene Family

In evaluating the evolution of the *ROS* network genes, the entire *ROS* network genes were aligned and phylogenetic tree constructed in order to understand the pattern and nature of their evolution. All the genes were classified into seventeen (17) different clades and each clade was made up of members of a particular family. The 17 clades were superoxide dismutase (SOD), ascorbate peroxidase (APX), monodehydroascorbate reductase (MDAR), dehydroascorbate reductase (DHAR), glutathione reductase (GR), catalase (CAT), glutathione peroxidase (GPX), ferritin (FER), respiratory burst oxidase homologue (RBOH), NADPH-like oxidase (FRO), Ubiquinol oxidase (AOX), peroxiredoxin (PrxR), thioredoxin reductase (Trx), Glutaredoxin (GLR), and peroxidase (PER) (Figure 2A). A total of 683 *ROS* network genes were found to have undergone duplications, in which 631 genes through segmental duplication while only 52 genes were tandemly duplicated (Appendix A). The results were in agreement to previous findings in which a number stress responsive genes evolution were majorly governed by segmental type of gene duplication, for instance the *LEA* genes [27]. Gene duplication is considered as the main processes which do lead to the expansion of various gene families in organisms, and it is a prevailing feature in plant genomes [28,29]. Generally, a high number of gene copies in a gene family are maintained through large-scale segmental duplication or small-scale tandem duplication during evolution.

Synteny analysis, showed a total of 161, 166, 99, and 105 orthologous genes synteny blocks between At and A, Dt and A, At and D, and Dt and D, respectively (Figure 2B–D). It is interesting to note that more *ROS* genes were observed between the syntenic blocks among the three cotton species, for instance, at most five genes were found within the syntenic regions between At and A, and the same number of genes were also observed between Dt and A. However, at most 20 genes have been found within the syntenic regions between Dt and D (Table 1 and Appendix A). The high number of *ROS* genes detected within the same syntenic regions showed that this syntenic regions have been highly conserved, and this indicates their integral role they play within the plants. This result supports the principle of asymmetric selection of genes within the genomes [8]. 

### 2.4. Orthologous ROS Gene Analysis and Evolution Prediction through Synonymous (K_s_) and Non-Synonymous (K_a_) Rate of Substituion

Allopolyploid cotton evolved following trans-oceanic dispersal of an A genome diploid to the Americas, where the immigrant underwent hybridization, as female, with a native D genome diploid similar to modern *G. raimondii* [30]. The formation of ROS network might be earlier than that of *Gossypium* diverged from its closest relatives and Allopolyploid cottons formation. Phylogenetic analysis (Appendix A) revealed that, a total of 241 paired genes, were orthologous among the three cotton species, *G. raimondii*, *G. arboreum* and *G. hirsutum*, found to be either directly or indirectly involved in the ROS network. In the analysis of the evolutionary history of the tetraploid cotton, 243 genes were orthologous between *G. arboreum* and *G. raimondii*. *G. arboreum* contained 14 genes, but not in *G. raimondii*, while only five specific genes were found to be orthologs between *G. arboreum* and At subgenome. In the D genome, *G. raimondii* had 17 genes, but not in *G. arboreum* and only five specific genes were found to be ortholog between *G. raimondii* and Dt subgenome (Figure 3A). This result suggested that 243 genes were formed evolved much earlier before the separation of the A and D cotton genomes, which are the *G. arboreum* and *G. raimondii*, respectively. Thirty-one (31) genes were found to have evolved before the separations of the two diploid cotton species, and even the emergence of the allotetraploid cotton. three genes were found to have originated from D, and introgressed into the Dt subgenome, of the allotetraploid cotton, while some element of gene loss was evident, this could have been occasioned by either loose of function or chromosomal rearrangement during the evolution of allotetraploid from the two diploid cotton species, A and D cotton genomes (Figure 3B). The synonymous substitution value (*K*_s_) peaks of orthologous gene pairs between At/A and Dt/D was estimated at 0.004 and 0.015, respectively. The distribution of 241 putative ortholog pairs between the three cotton species/genomes (mode *K*_s_ = 0.0040–0.015) clearly indicated that the secondary peak for cotton, lies to the left of the ortholog peak, which represents a burst of gene duplications that occurred in cotton after the separation between the two lineages. Similar observation has been made in the analysis of the genome evolution of soybean, which showed that soybean genome also underwent whole genome duplication [31]. From our findings, the *ROS* genes network seemed to have occurred in tandem with the evolution of tetraploid cotton, approximately 0.8–2.8 million years ago (MYA) [2]. Both synonymous site (*K*_s_) peaks of the ROS orthologous gene pairs between the two cotton progenitors, A and D genomes, and two subgenomes (At and Dt) was 0.045 (Figure 3C,D). Moreover, we estimated the *ROS* genes network divergence time between the A and D genomes to have occurred approximately 8.6 MYA, in the results was in agreement to the divergence time of the two diploid cotton, *G. arboreum* and *G. raimondii* which occurred 5–10 MYA [32]. Additionally, the synonymous (*K*_s_) and non-synonymous (*K*_a_) substitution values ratio (*K*_a_/*K*_s_) were less than one (1) that suggested that the *ROS* gene network evolution, was primarily governed by purifying selection, similar results were also obtained in the analysis of the 156 paralogous pairs of the *LEA* genes in cotton, in which over 99% of the orthologous gene pairs had the *K*_a_/*K*_s_ ratio values of less than 1 [27]. 

### 2.5. Analysis of the Expression Levels of Homeolog and Ortholog Genes ROS Network Genes at 10 DPA and 20 DPA of Cotton Fiber Development

RNA-seq data of the public database (https://www.ncbi.nlm.nih.gov/) were downloaded and reanalyzed, which showed 192 *ROS* genes pairs were expressed during cotton fiber development (Appendix A). Moreover, we analyzed the homeolog and ortholog gene pair’s expression levels at 10 and 20 DPA in the three cotton species, *G. hirsutum*, *G. raimondii*, and *G. arboreum*. It is interesting that the expression levels of genes encoding the SOD proteins, such as the CSD, FSD, and MSD proteins exhibited lower expression levels at 10 and 20 DPA in *G. raimondii* but were significantly higher in *G. arboreum* and *G. hirsutum*. The expression levels of most of the *SOD* genes were higher at 10 DPA than at 20 DPA (Figure 4A,B). The result showed that the *SOD* genes are integral at 10 DPA, being a critical stage of fiber initiation and elongation, moreover, at the stage of 10 DPA, ABA inhibits fiber cell initiation and elongation [33]. Moreover, previous studies have shown that cotton annexin proteins; AnxGh1:AAR13288, AnxGh2:AAB67993, AnxGhFx:FJ415173, and AnxGhF:AAC33305 are present in higher amount in fibers of 10 DPA wild-type plants as compared to the fuzzless-lintless mutant [34]. This result indicated that *SOD* genes were highly expressed in taxa with long fiber (*G. arboreum* and *G. hirsutum*), while were either down regulated or partially expressed in cotton species with shorter fibers, such *G. raimondii* of the D genome. Additionally, the expression level of *SOD* genes showed up regulation at fiber elongation phase (10 DPA) compared to the secondary cell wall biosynthesis/thickening phase (20 DPA). The results obtained for the expression pattern of the *ROS* genes, correlated positively to previous findings on the expression patterns of some of the genes known to be highly involved in fiber development in cotton, such as APX, MDAR, GR, GPX, FER, AOX, PrxR, Trx, GLR, and PER [21,22,35]. The *SOD* gene family and other ROS scavenging genes are believed to be involved in cotton fiber development through the change in their expression level in the course of fiber development period. On the other hand, most of the genes with functional role in the regulation of ROS, including *RBOH* and *FRO* genes were up regulated at the secondary cell wall biosynthesis phase (20 DPA) relative to fiber elongation phase (10 DPA) in *G. hirsutum*, *G. arboreum*, or *G. raimondii*. In summary, ROS level might be low at fiber elongation phase, but high at secondary cell wall biosynthesis phase in *G. hirsutum* and *G. arboreum*. Compared with *G. hirsutum* and *G. arboreum*, although the expression model of ROS production genes was similar in *G. raimondii*, the expression level of ROS removing genes was lower. Fine regulation of steady-state levels of ROS are essential for proper cell elongation [36], and long and short fiber properties were regulated by ROS removing genes, such as *SOD* gene family. Moreover, the Ascorbate peroxidase 1 *(APX1)*, has been found to be integral in the process of cotton fiber development, and that *APX* gene profiling in upland cotton, *G. hirsutum* has provided valuable information on the redox homeostasis in the process of fiber development [37]. Furthermore, when the cytosolic H_2_O_2_-scavenging enzyme, APX1, the whole chloroplastic H_2_O_2_-scavenging process stops in *A. thaliana* and the level H_2_O_2_ increases resulting into protein oxidation [38]. Ferric Chelate Reductase-2 (*FRO2*), Ferric Chelate Reductase-5 (*FRO5*), and respiratory burst oxidase homologs (*RBOH6*) orthologous gene pairs, modulation ROS level in fiber development, were identified by GWAS analysis and RNA-seq analysis [8]. Furthermore, the RBOH do modulate the ROS and in turn promote root growth in Arabidopsis [39].

### 2.6. ROS Network Genes Response to Abiotic Stress

We downloaded data of ROS-gene expression levels from pubic RNA-Seq data of *G. hirsutum* (TM-1). After analyzing gene expression under cold, heat, dehydration, and salt stress treatment, a total of 135 *ROS* genes were founded to be upregulated by abiotic stresses (Appendix A). Compared with the controls, 92 (68%), 79 (59%), 74 (55%), and 85 (62.9%) genes were differently expressed under cold, heat, dehydration, and salt stress, respectively. Twenty-nine genes were differentially expressed under all four different stress conditions, which included CSD, APX, MDAR, CAT, GPX, FER, RBOH, FRO, AOX, GLR, and PER. The majority of the differentially expressed genes (DEGs) showed more expression changes under cold stress than under other abiotic stresses, suggesting that *ROS* genes were more sensitive to cold stress. ROS-producing genes were up regulated after cold, heat, dehydration, and salt treatments. FRO5 gene shown higher expression level than other ROS-producing DEGs and was upregulated after 3 and 6 h of post stress exposure. *APX1* and *MDAR1* genes, showed higher expression level than other ROS-scavenging DEGs, and were upregulated after 3 and 6 h (Figure 5). These results proved that the *ROS* gene do respond positively towards abiotic stress during the early developmental stages. *RBOH6* and *APX1* genes were highly expressed in *G. hirsutum* D subgenome than At subgenome which suggested that some gene pairs of *G. hirsutum* D subgenome and A subgenome might be playing different roles in cotton growth and development [8].

### 2.7. Analysis of Morphological and Physiological Changes under Salt Stress

Four semi-wild (Latifolium 130, Latifolium 32, Latifolium 40, and Marie-galante 85) and two cultivated (CRI12 and CRI16) *Gossypium hirsutum* accessions were evaluated for phenotypic traits under salt treatment. Among them, Marie-galante 85 (M85) was highly tolerant to salinity stress, as evidenced by significantly higher relative stem length, relative root water content, relative stem water content, and lower rate of area of damaged leaves (Table 2; Figure 6). Compared with the controls, proline content was significantly enhanced under salt stress in M85, but was significantly lower in CRI12. These results suggested that M85 is highly tolerant compared to CRI12, to salt stress. The content of MDA was not significantly changed between normal condition and salt stress condition in both CRI12 and M85. However, the concentration level of MDA was relatively lower in the leaf tissues of M85 compared to the level in CRI12. Proline plays an important role in maintaining osmotic balance and protecting some important enzymes in plant cells, thus mainly acts as osmoprotectant when plants are exposed to water deficit condition, the detection of high proline content in the leaf tissues of M85, showed that the genotypes has higher adaptability to salinity compared to CRI12. Therefore, the same with salt (NaCl) stress, cotton is exposed to complex alkali-salt stress could result in osmotic stress. Malondialdehyde (MDA) is the final product of lipid peroxidation, and its content can reflect stress tolerance level of the plant [36]. However, in our study, the content of MDA was not significantly changed after salinity treatment. It suggested that alkali-salt stress has not resulted in lipid peroxidation of M85 and CRI12. Taken together, cotton regulation of complex alkali-salt stress was different from salt (NaCl) stress, and M85 had a higher tolerance to osmotic stress than CRI12. Reactive oxygen species (ROS) balance was also observed. The content of hydrogen peroxide (H_2_O_2_) and the activity of superoxide dismutase (SOD), peroxidase (POD), catalase (CAT), ascorbate peroxidase (APX), and glutathione reductase (GR) were measured. The content of hydrogen peroxide was significantly enhanced in root and leaf of CRI12 under alkali-salt stress. Hydrogen peroxide was also increased in leaf of M85, but significantly decreased in root of M85 under alkali-salt stress. Furthermore, we found that the activity of POD, CAT, APX, and GR was not significantly changed in roots of M85, suggesting that ROS production and removal was balanced in roots of M85. The content of hydrogen peroxide was higher in leaf than that in root. In addition, the activity of POD was significantly increased in leaf of M85, but POD, CAT, APX, GR, and SOD was not significantly affected. This result suggests that the leaf of M85 was better to remove excessive accumulation of ROS. Additionally, the activity of SOD was significantly increased under alkali-salt stress in root of M85, and the activity of POD and CAT was significantly enhanced in root of CRI12, suggesting that redox enzymes was act on different conditions, flexibly [40].

### 2.8. RNA-Seq Analysis and ROS Network Response to Alkali-Salt Stress

After Illumina Hiseq2500 sequencing, a total of 39.62–57.16 M clean reads were generated for each sample, in which approximately 85.65%–89.61% of reads were mapped to the reference genome, and 77.79%–82.27% of clean reads were mapped to the unique locations of the reference genome (Appendix A). Unique mapped reads were assembled and annotated with Cufflinks and were used for analyzing mechanism of plant response to alkali-salt stress. Based on a cutoff *q*-value ≤ 0.01 and fold change ≥ 2, different expressed genes (DEGs) were identified. The number of DEGs was higher at 12 h post alkali-salt stress treatment than that of 3 h in both roots and leaves in CRI12 and M85 moreover; the ROS network genes analysis in the same region by principal component analysis showed genes were aggregated at a particular region (Figure 7A,B). 

The ROS balance system was found to highly enriched, the finding which was consistent with the previous studies [41]. Moreover, the majority of the DEGs were found to be enriched in response to oxidative stress (GO0006979), oxidation-reduction process (GO0055114), oxidoreductase activity (GO0016717, GO0016491, and GO0046857), and peroxidase activities (GO0004601) (Appendix A). We also observed H_2_O_2_ accumulation in leaves and roots under alkali-saline stress. The result shows the importance of ROS networks in acclimation for alkali-saline stress in *Gossypium hirsutum*. DEGs were also enriched in protein serine/threonine kinase activity (GO0004674). This result supports that the serine/threonine kinases, act as one of signal transduction molecular, is an essential kinase for oxidative stress responses [42]. Genes of ROS network that control ROS levels, encoded most oxidation-reduction proteins. The correlation analysis (*R*^2^ > 0.7) between different cotton accessions through orthologous genes expression levels analysis were significantly higher (Figure 7C). Based on the analysis of the various *ROS* network genes, we further predicted the interlink between fiber and abiotic stress responsive *ROS* genes (Figure 8). We found that CSD1-6, FSD1-2, MSD 1-2, APX3, APX11, FRO5, FRO6, and RBOH6 have a functional role in enhancing fiber development, more specifically at 10 and 20 DPA, while CSD1, APX1, APX2, MDAR1, GPX4-6-7, FER2, RBOH6, RBOH11, and FRO5 are specifically involved in enhancing abiotic stress tolerance, more salt stress. Finally, ten genes were selected from three different libraries and validated through RT-qPCR, the expression levels were same as per the RNA sequencing data (Figure 7D).

## 3. Discussion

In plants, many different biology processes are controlled by ROS [43]. For example, fiber development and biotic and abiotic stresses have previously been found to be partially regulated by ROS in cotton [44,45]. During 5–10 MYA, *Gossypium* split into three major diploid lineages: the New World clade (D genome), the African-Asian (A, B, E, and F genomes) clade, and the Australian clade (C, G, and K genomes) [1]. Allotetraploid Upland cotton was formed approximately 1–2 MYA [46] and domesticated for at least 4000–5000 years ago [47]. During human domestication history, the central focus was the fiber quality and quantity, but in the recent past, researchers have found that fiber and fiber qualities are not only affected by the intrinsic nature of the cotton plant, but greatly influenced by the environmental conditions, and therefore in elucidating the mechanisms of fiber development and improvement of its qualities, multidimensional approach has so far been preferred by plant breeders. The change in environmental condition, occasioned by continued human interference, rainfall has become scarce and very erratic in nature, in addition to the emergence of other forms of abiotic stresses, such as salinity, heat, and cold, among others [48]. The sessile nature of plants, makes them to be much prawn to full effects of abiotic stresses, and thus in the course of plant evolution, various survival strategies have been adopted by plants, more so the evolution of various stress responsive plants transcription factors, in which *ROS* genes are among one of them. In our study, we found that ROS network was conserved in cotton. The Evolutionary history of *ROS* genes network in cotton was similar to *Gossypium* evolution. Thus, ROS network was positively selected during cotton evolution and the result of *K*_a_/*K*_s_ analysis also supported this view. Gene order and colinearity of allotetraploid cotton are largely conserved between the A and D subgenomes and the extant D progenitor genome (*G. raimondii*) [47], and the genes colinearity of D09 (*G. raimondii*) was conserved with Dt05 and At05 (*G. hirsutum*). Interestingly, more potential *ROS* genes were mapped to A10 (35, 12.36%), D09 (40, 13.89%), At05 (46, 18.11%), and Dt05 (38, 14.28%) in *G. arboreum*, G. raimondii, *G. hirsutum* A subgenome, and *G. hirsutum* D subgenome, respectively. It suggested that the location of *ROS* genes was conserved during the domestication process and D09, Dt05, and At05 play a core role in regulated ROS dynamic balance. Above all, ROS plays an important role in cotton growth and development, because of ROS network genes were positive selection and conserved during evolution and domestication of cotton.

At least 15 gene families (SOD, APX, MDAR, DHAR, GR, CAT, GPX, FER, RBOH, FRO, AOX, PrxR, Trx, GLR, and PER) of the ROS network were identified in *Gossypium*. SOD, APX, and *GPX* gene families were identified and analyzed. Our results are consistent with the results of *SOD* gene analyses in previous studies [37] and *SOD* genes cover the chromosome at the same locations. However, two genes were annotated as copper chaperone for superoxide dismutase in *G. arboreum* and *G. raimondii*. Compared with a previous study, the member of *GPX* genes family was different because of different genomes (NBI_*Gossypium hirsutum* vs. BGI_*Gossypium hirsutum*) used for identifying *GPX* genes. In previous studies, ROS were found to regulate development, differentiation, redox levels, stress signaling, interactions with other organisms, systemic responses, and cell death in higher plants [49,50,51]. Wild cottons allocate greater resources to stress response pathways, while domestication led to reprogrammed resource allocation toward increased fiber growth, possibly through modulating stress-response network [52]. Modulating reactive oxygen species (ROS) production and scavenging could be regulating cotton fiber elongation [53], abiotic stress [41].

In previous studies, different developmental or environmental signals feed into the ROS signaling network and perturb ROS homeostasis in a compartment-specific or even cell-specific manner [12]. Then, the fluctuant ROS is perceived by receptor proteins (NPR, HSF), redox-sensitive transcription factors, or direct inhibition of phosphatases [54]. Serine/threonine protein kinase (OXI1) is involved in ROS sensing and the activation of mitogen-activated-protein kinases (MAPKs) [55] that plays a central role in ROS signal network [56,57,58]. ROS-producing or -scavenging pathway is activated to regulate ROS levels. Different developmental, metabolic and defense pathways are activated to regulate growth and development. In cotton, several families of transcription factors, such as *MYB*, *WRKY*, *AP2*/*ERF*, and *NAC* have been found to be involved in regulating plant response to abiotic stress [59], and some of them are also involved in ROS signal network [60].

Cotton fiber development is a complicated biological process. Many researches demonstrate that fiber development was regulated by ROS-mediated pathway. In the previous study, 119 association signals, 71 SNP associated with yield-related traits, 45 association with fiber qualities, and three associated with resistance to Verticillium wilt, were identified through genome-wide association study, and a total of 43 *ROS* genes were found [61]. Among the 43 genes, seven genes were within the associated SNP signals region of 200 kb (±100 kb), and one of the 43 genes, the following genes were found to have a direct involvement in the regulation of fiber development, for instance *Gh_A13G1654 (GLR34)* was at the downstream, and was highly associated to signals A13:75463332 (associated with fiber elongation), A13:75463913 (associated with fiber strength), and A13:75473149 (associated with fiber length); *Gh_D05G2413 (FRO2)* was at the downstream, at the boll weight association signal D05: 24,174,044 (associated with boll weight). This finding may partially support the speculation that ROS network associated with some different plant traits. ROS is involved in regulating plant cell expansion [62], which suggests ROS may also be involved in regulating cotton fiber development [17]. High concentration of ROS in wild-cotton fiber cells inhibits fiber elongation by regulating secondary-cell-wall synthesis [8]. Moreover, the H_2_O_2_, involved in cell-wall relaxation, is necessary molecule for cell elongation, however, higher ROS levels may halt cell elongation [63,64,65]. Compared with fiber elongation (10 DPA), ROS level was regulated by ROS network, was upregulated at secondary cell wall biosynthesis (20 DPA) [21]. It is interesting that the expression levels of the *SOD* genes and some other ROS-scavenging genes were higher *G. hirsutum* and *G. arboreum*, and they are known to have relatively longer fiber compared *G. raimondii*. Moreover, *G. hirsutum* and *G. arboreum*, exhibited lower ROS-scavenging enzymes, which implied that they had the higher ROS-producing ability than *G. raimondii*. The results therefore showed that perturbed ROS levels as a result of the *SOD* genes and some other genes could perhaps have a net effect on the regulation of various stages of fiber development in *G. hirsutum*, *G. arboreum,* and *G. raimondii*. In addition, fluctuating levels in ROS levels, caused by different expression level of ROS-scavenging genes and ROS-producing genes at fiber elongation and secondary cell wall biosynthesis phases, showed the integral role played by the *ROS* genes in fiber development. 

Almost all the abiotic stress factors do lead to the development of secondary stress condition such as osmotic and oxidative stresses [66]. RNA-seq data, physiological, and phenotypic index analysis in the two upland cotton genotypes, M85 and CRI12 suggested that the *ROS* gene network is partially responsible to salinity stress tolerance. The majority of the *ROS* genes were expressed under abiotic stress, which demonstrated that ROS network was also interconnected with abiotic stress tolerance in cotton [67,68,69]. According to the expression values (FPKM) of the ROS network genes in *G. hirsutum* TM-1, the expression levels of the homologous genes in under various abiotic stress conditions were: 92 (68%), 79 (59%), 74 (55%), and 85 (62.9%) under cold, heat, dehydration, and salt stress conditions, respectively. A total of 29 genes were differentially expressed under four different stress conditions, in which out of the six genes, three genes were the ROS-producing genes, while the other three were the ROS-scavenging genes. 

## 4. Materials and Methods 

### 4.1. Identification of the ROS Genes in Cotton

By against the complete genomes of *Gossypium hirsutum* (NAU-NBI Assembly v1.1 & Annotation v1.1), *G. arboreum* (BGI Assembly v2.0 & Annotation v1.0), and *G. raimondii* (JGI assembly v2.0 & annotation v2.1) which were downloaded from CottonGen database (https://www.cottongen.org), a total of 152 *ROS* genes controlling ROS levels in *Arabidopsis* [12] were used to identify *ROS* genes in cotton genome using BLASTP using the following criteria [70]: the longest transcript in each gene loci was chosen to represent that locus and the sequences were filtered out when (1) CDS length with < 150 bp, (2) CDSs with percentages of ambiguous nucleotides (‘N’) > 50%, (3) CDSs with internal termination codons, and (4) the CDSs with hits (BLAST identities ≥80%) to RepBase sequences (http://www.girinst.org/repbase/index.html). The conservative domains of candidate genes were obtained from Pfam website (http://pfam.xfam.org/) [71] and NCBI Web CD-Search Tool (https://www.ncbi.nlm.nih.gov/Structure/bwrpsb/bwrpsb.cgi) [72]. The biophysical characters of the encoded proteins were computed using the ExPASy ProtParam tool (http://us.expasy.org/tools/protparam.html).

### 4.2. Phylogenetic Tree, Orthologous Gene Pairs and Values of K_a_, K_s_ and K_a_/K_s_

Full-length sequences of ROS-network proteins were first aligned with the ClustalX program (http://www.clustal.org/clustal2/) [73], then phylogenetic trees were constructed by using neighbor-joining (NJ) method with 1,000 bootstrap replicas [74] and the Poisson model by using MEGA 7.0 software (http://www.megasoftware.net). Meanwhile, the orthologous gene pairs of the ROS network in A, D genomes, At and Dt subgenomes were searched by InParanoid software (http://inparanoid.sbc.su.se/cgi-bin/index.cgi). Additionally, the evolutionary rates *K*_a_, *K*_s_, and *K*_a_/*K*_s_ ratio were estimated by *K*_a_*K*_s__Calculator package (https://kakscalculator.herokuapp.com/). On the basis of the synonymous substitutions per year (λ) of 2.6 × 10^−9^ for cotton, the divergent time of ROS orthologous gene pairs were estimated by T = *K*_s_/2λ × 10^−6^ MYA [47]. 

### 4.3. Chromosomal Locations, Conserved Synteny Blocks and Gene Duplication

Chromosomal positions of the *ROS* genes were obtained from the annotation information of cotton genomes (*G. arboreum*, *G. raimondii* and *G. hirsutum*). The chromosomal distributions of genes were drafted on the cotton chromosomes according to the gene positions by Mapchart software [75]. The conserved synteny blocks between A genome and At subgenome, D genome and Dt genome in cotton were inferred using the MCSCANX program with the default parameters (http://chibba.pgml.uga.edu/mcscan2/) [76]. A local BLASTP program was run on different genomes with parameters (*e*^−10^). The syntenic relationships were illustrated with the CIRCOS program (http://circos.ca/) [77]. The duplicate gene classifier program of MCSCANX software was used to identify each ROS network gene duplication with the default parameters (OVERLAP WINDOW: 5). All the ROS network genes were classified into various types of duplications.

### 4.4. Plant Materials and Salt Stress Treatment

*G. hirsutum* var. marie-galante 85 (M85) and *G. hirsutum* cv. CRI12 were selected for the salinity treatment. Semi-wild cotton marie-galante was originally collected from Yucatan, Mexico and cultivated in the National Wild Cotton Nursery (Sanya, Hainan, China). *G. hirsutum* cv. CRI12 was widely planted in china. The seeds of M85 and CRI12 were first germinated at 28 °C in a 16 h light/8 h dark cycle. Then, seedlings were planted in the normal solutions for three weeks. The similar growing seedlings were selected for salt-alkali stress. The salt-alkali treatment was imposed by simulating the actual saline condition of Xinjiang province, a region commonly known for salt stress [78]. We examined the soil salt composition and simulated a similar environment to impose salt-alkali stress, the composition was made of 2.664 mg:g^−1^ of calcium chloride (CaCl_2_), 0.179 mg:g^−1^ of sodium hydrogen carbonate (NaHCO_3_), 9.94 mg:g^−1^ of sodium sulphate (Na_2_SO_4_), 0.848 mg:g^−1^ of potassium sulphate (K_2_SO_4_) and finally 3.587 mg:g^−1^ of hydrated magnesium sulphate (MgSO_4_·7H_2_O) [79]. 

### 4.5. Measurement of Morphological, Physiological and Biochemical Traits

Samples of roots and leaves that collected at 48 h after salt-alkali stress treatment were used for measuring physiological traits. And, seedlings were also used for measuring phenotype at 48 h after salt-alkali stress treatment. Malondialdehyde (MDA) and Proline (PRO), the most important protective macromolecules in response to salt stress, were measured using the kits (Nanjing Jiancheng Bioengineering Institute). Hydrogen peroxide (H_2_O_2_), superoxide dismutase (SOD), peroxidase (POD), ascorbate peroxidase (APX), catalase (CAT) and glutathione reductase (GR), important ROS and ROS-scavenging molecules, were measured using the corresponding assay kits (Nanjing Jiancheng Bioengineering Institute). The root length (RL), stem length (SL), roots fresh weight (RWC) and stem fresh weight (SWC) were measured after 48 h salt stress. The value of SPAD, one of photosynthesis indices, was measured using chlorophyll meter model SPAD-502.

### 4.6. RNA Extraction, cDNA Library Construction, and RNA-Seq

Roots and leaves were collected at 0, 3, 12 and 48 h time points after salt-alkali stress treatment. Then, collected samples were used for transcriptome sequencing. Total RNA was extracted from each cotton sample using TRlzol Reagent (Life technologies, CA, USA) according to the instruction manual. RNA integrity and concentration were checked using an Agilent 2100 Bioanalyzer (Agilent Technologies, Inc., Santa Clara, CA, USA). The mRNAs were isolated by NEBNext Poly (A) mRNA Magnetic Isolation Module (NEB, E7490) by New England BioLabs (Ipswich, MA, USA). The cDNA libraries were constructed by following the manufacturer’s instructions of NEBNext Ultra RNA Library Prep Kit for Illumina (NEB, E7530) and NEBNext Multiplex Oligos for Illumina (NEB, E7500) (Illumina, San Diego, CA, USA). Briefly, the enriched mRNA was fragmented into RNAs with approximately 200 nt, which were used to synthesize the first-strand cDNA and then the second cDNA. The double-stranded cDNAs were performed end-repair/dA-tail and adaptor ligation. The suitable fragments were isolated by Agencourt AMPure XP beads (Beckman Coulter, Inc., Indianapolis, IN, USA) and enriched by PCR amplification. Finally, the constructed cDNA libraries were sequenced on a flow cell using an Illumina HiSeq™ 2500 sequencing platform.

### 4.7. Expression Analysis

Whole-transcriptome sequencing data for *G. hirsutum* were obtained from the NCBI Sequence Read Archive (SRA) (http://www.ncbi.nlm.nih.gov/sra) to analyze the tissue/organ-specific, stage-specific and stress-induced expression patterns of cotton *ROS* genes [47]. The details of the SRA are shown in (Appendix A). The row data of RNA-seq of M85 and CRI12 were separately analyzed and the clean reads were obtained by removing reads containing adapter, reads containing ploy-N and lower quality reads from raw data. At the same time, Q_30_ [80], GC-content and sequence duplication level of the clean data were calculated. Raw sequences were transformed into clean reads after data processing. These clean reads were then mapped to the reference genome sequence. Only reads with a perfect match or one mismatch were further analyzed and annotated based on the reference genome. Tophat2 [81] software was used to map with reference genome. Base on the reference genome, using Cufflinks software [82], mapped reads had been assembled. Quantification of gene expression levels was estimated by fragments per kilobase of transcript per million fragments mapped (FPKM). Base on mapped reads, using FPKM as the index, each gene was estimated by Cuffquant and Cufform. EBSeq software was used to identify the differential expression genes by Fold change ≥ 2 and FDR ≤ 0.01 (False Discovery Rate). FDR was corrected using Benjamini-Hochberg method by p-value. Finally based on the RNA sequence expression data, ten (10) genes were selected and their expression validated through RT-qPCR in relation to salt-alkali stress. In each, three libraries were chosen and the gene specific primers were designed by using primer premier 5 (Appendix A). The RT-qPCR analysis was carried out as described by Magwanga et al [83], with GhActin as the internal control gene.

### 4.8. Statistical Analysis

The experimental data on phenotype and physiology were statistically analyzed using the SAS version 8.2. Graphic presentations were performed using OrginPro 8.0 program and R packages.

## 5. Conclusions

Due to the ever-changing environmental conditions, the available land for cultivation is declining as a result of various environmental stress factors such as drought and salinity, among others [84]. The combined effect of drought and salinity has been estimated to cause huge losses in agricultural crops as over 6% of arable lands are saline [85]. Thus non edible crops such as cotton are currently being grown in harsher conditions, in soils with high levels of salt content, but due to the importance of the cotton plant, focus is aimed of developing a more robust and salt tolerant cotton cultivars. When plants are exposed to any form stress, the ROS equilibrium shifts leading to high concentrations which results in toxicity and eventually leads to total plant death [86]. In this research work, we carried out deep transcriptome analysis of the ROS network genes, evaluated their evolution pattern and their expression levels under abiotic stress and fiber development stages. A total of 515, 260, and 261 ROS candidate proteins were identified in *G. hirsutum*, *G. arboreum*, and *G. raimondii*, respectively. The high numbers of the ROS network genes were found to have evolved through segmental gene duplication compared to tandem, a result which was in agreement to evolution pattern of various stress responsive genes such the *LEA* genes which evolved mainly through segmental type of gene duplication [27]. Moreover, RNA sequencing and RT-qPCR validation revealed that the ROS genes had a putative role in enhancing abiotic stress tolerance in cotton and in turn promote fiber development in cotton. In relation to fiber development, the following genes were found to be highly upregulated, CSD1,2,3,5,6; FSD1,2; MSD1,2; APX3,11; FRO5.6; and RBOH6 while for salt stress tolerance, CSD1, APX1, APX2, MDAR1, GPX4-6-7, FER2, RBOH6, RBOH11; and FRO5 were found to have a significant role. This research lays a significant foundation for future exploration of the *ROS* genes in developing a more stress tolerant cotton genotypes and with superior fiber quality. 

## Figures and Tables

**Figure 1 ijms-20-01863-f001:**
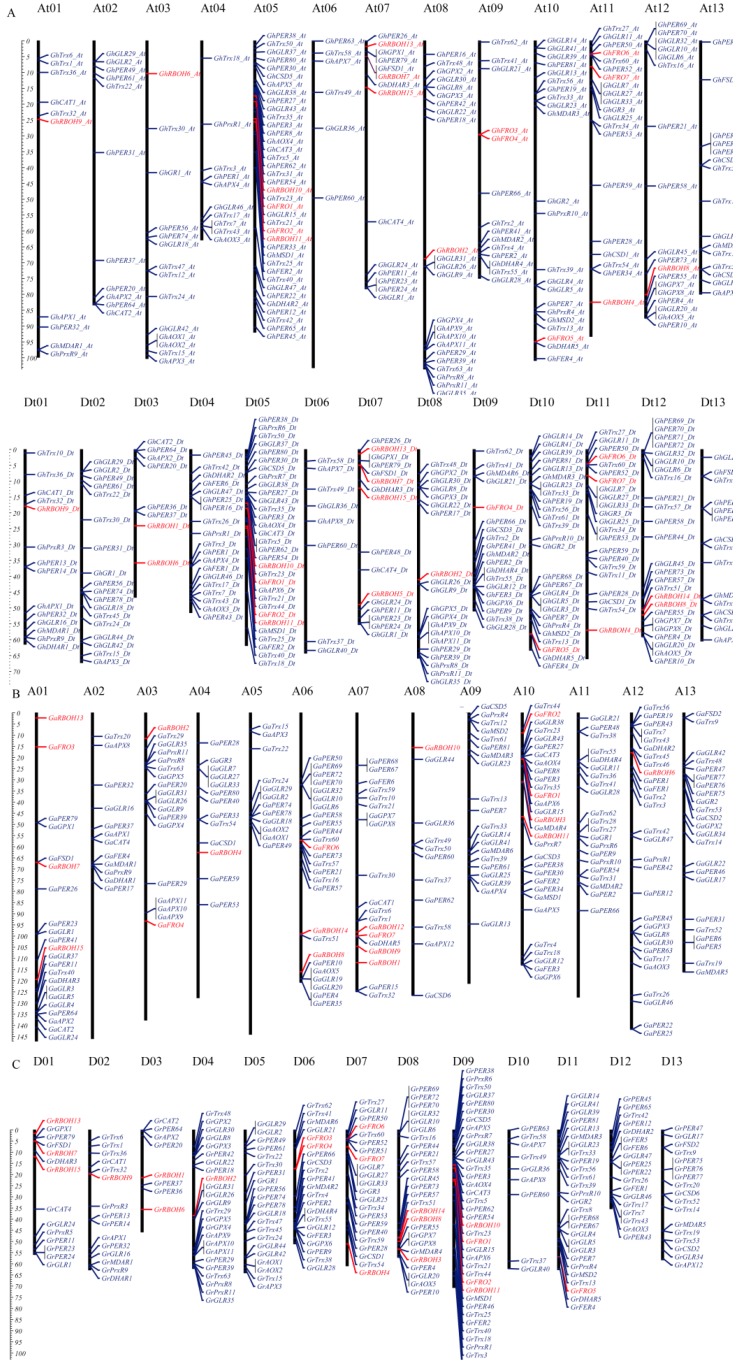
Distribution of reactive oxygen species (ROS) network genes in chromosomes of *G. hirsutum*, *G. arboreum* and *G. raimondii*. (**A**) Distribution of ROS network genes in chromosomes of *G. arboreum*. (**B**) Distribution of ROS network genes in chromosomes of *raimondii*. (**C**) Distribution of ROS network genes in chromosomes of *G. hirsutum*.

**Figure 2 ijms-20-01863-f002:**
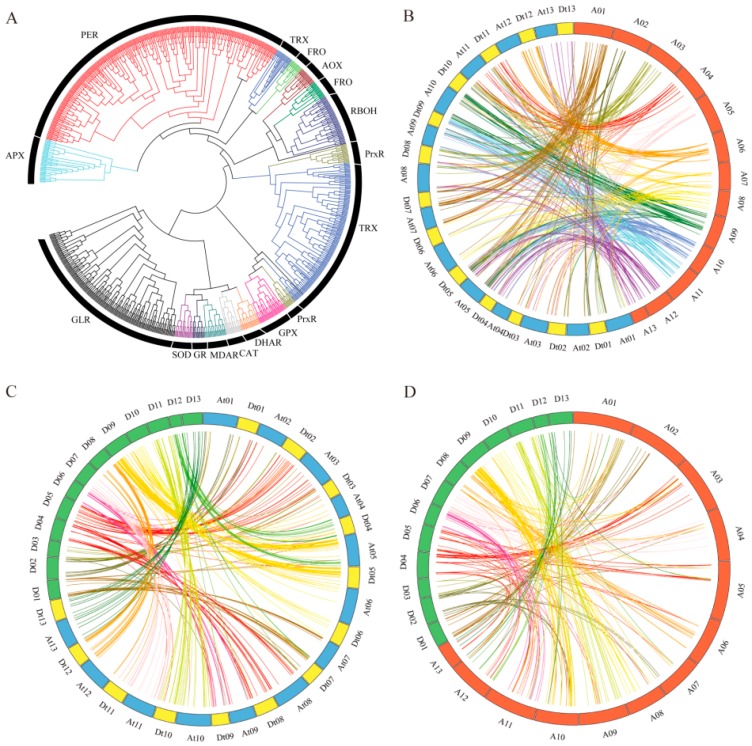
Phylogenetic tree and synteny analysis. (**A**) Phylogenetic tress analysis of all the cotton *ROS* network genes in *G. hirsutum*, *G. raimondii*, and *G. arboreum*. (**B**) Synteny analysis between A genome and allotetraploid AD_1_ genome. (**C**) Synteny analysis between D genome and allotetraploid AD_1_ genome. (**D**) Synteny analysis between A and D genomes. Red, green, yellow, and blue block means A genome, D genome, Allotetraploid cotton (TM-1) A and D subgenomes, respectively. The lines with different color mean genes of different chromosome (A and D genomes).

**Figure 3 ijms-20-01863-f003:**
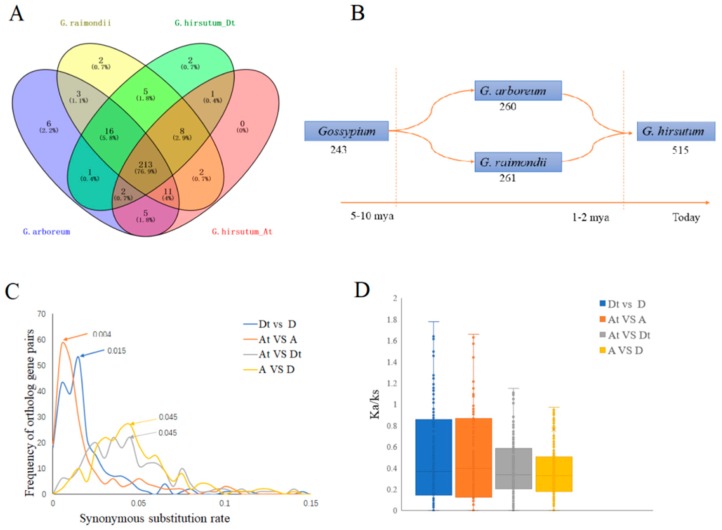
Orthologous gene pairs and evolution analysis of the *ROS* genes network in *G. arboreum*, *G. raimondii*, and *G. hirsutum*. (**A**) Orthologous gene pairs of the *ROS* genes network. (**B**) The orthologous genes pairs changes with a temporal depiction of phenomena that characterize polyploid evolution in Gossypium. (**C**) Distribution of *K*_s_ values for orthologous genes pairs between the A genome (*G. arboreum*), D genome (*G. raimondii*), and two subgenomes of TM-1(*G. hirsutum* At subgenome and Dt subgenome). At Dt;‘ t’ indicates tetraploid. Peak values for each comparison are indicated with arrows. (**D**) Distribution of *K*_a_/*K*_s_ value between the A genome, D genome and two subgenomes of TM-1.

**Figure 4 ijms-20-01863-f004:**
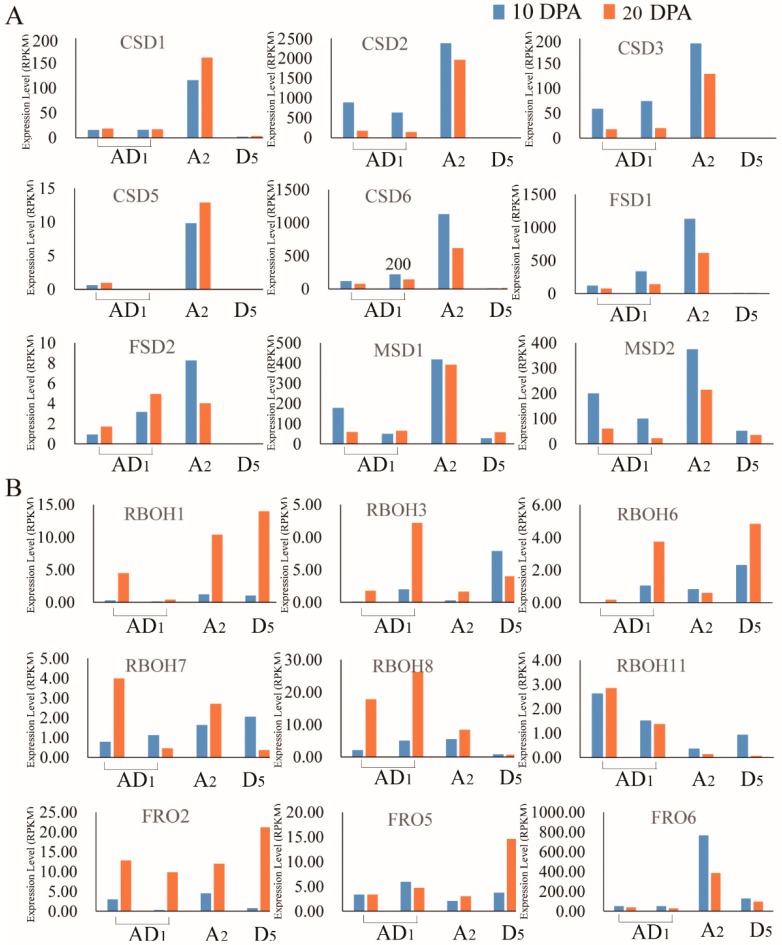
Expression level obtains from RNA-seq data and was show with RPKM (Reads per Kilobase of exon model per Million mapped reads). (**A**) The expression level of superoxide dismutase (SOD) genes family. (**B**) The expression of respiratory burst oxidase homologue (RBOH)and NADPH-like oxidase (FRO).

**Figure 5 ijms-20-01863-f005:**
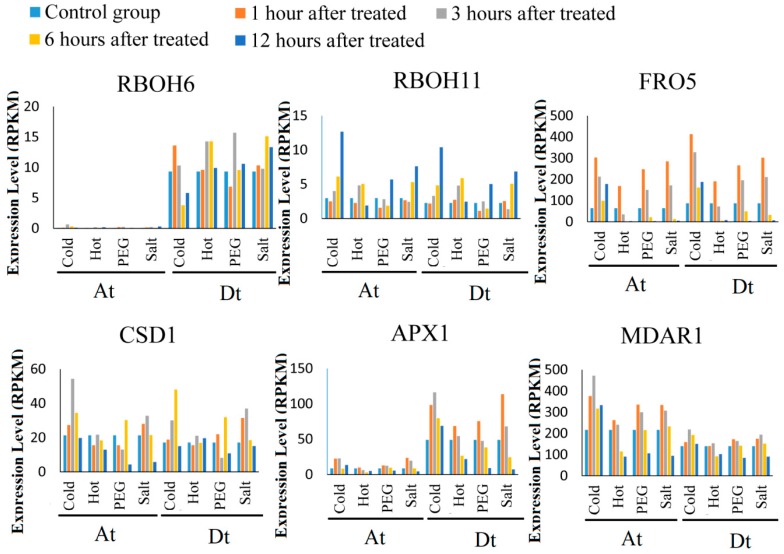
Expression of ROS Network genes in *G. arboreum*, *G. hirsutum*, and *G. raimondii* under abiotic stress. Expression level obtained from RNA-seq data and was show with RPKM (Reads per Kilobase of exon model per Million mapped reads). (**A**) The expression of *RBOH6/RBOH11* and *FRO5* orthologous gene pairs. (**B**) The expression level of *CSD1*, *APX1*, *and MDAR1* orthologous gene pairs.

**Figure 6 ijms-20-01863-f006:**
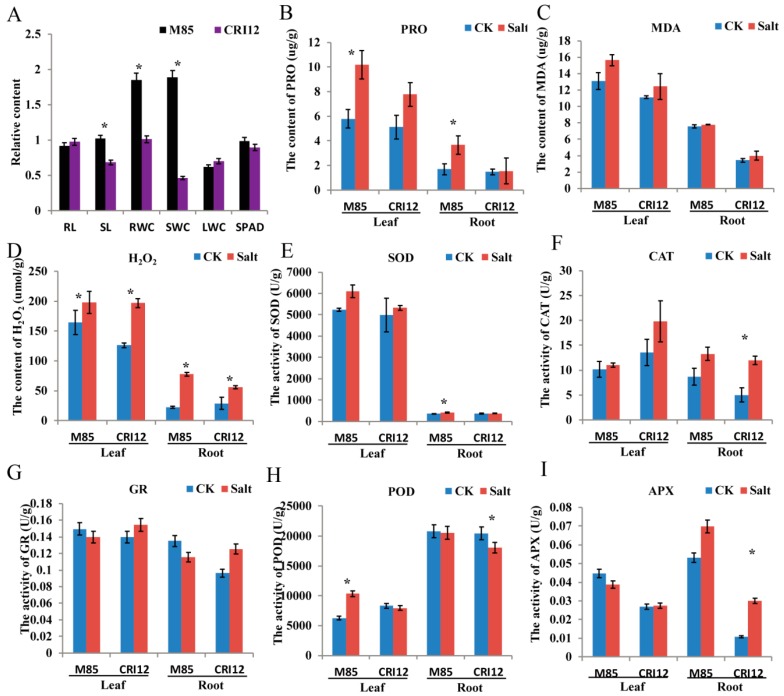
Complex alkali-salinity stress effects on the growth and the balance between intracellular active oxygen production and the defense system of cotton seedlings. (**A**) The phenotype of M85 and CRI12 indicated by blue and red arrows, respectively. The relative content of RL (root length), SL (stem length), RWC (root water content), SWC (stem water content), LWC (leaf water content), and SPAD (chlorophyll SPAD value) could visually show the variations among the plants. (**B**,**C**) The content of MDA (malondialdehyde) and PRO (proline) were measured at 48 h post to complex alkali-saline stress. (**D**) The content of hydrogen peroxide (H_2_O_2_) in leaves. (**E**–**I**) The enzymatic activity of superoxide dismutase (SOD), peroxidase (POD), catalase (CAT), ascorbate peroxidase (APX), and glutathione reductase (GR) were measured in leaves and roots, respectively. Error bars represent the standard deviation (SD) of three biological replicates.

**Figure 7 ijms-20-01863-f007:**
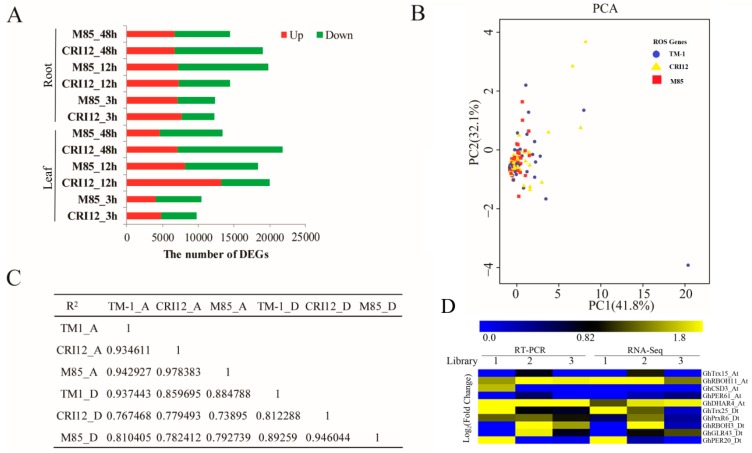
RNA-Seq analyses and RT-qPCR validation of the ROS network genes in M85 and CRI12 under alkali-salt stress. (**A**) The number of differentially expressed genes (DEGs) of *G. hirsutum races marie-galant* (M85) and *G. hirsutum cultivar* (CRI12) under complex alkali-saline stress at 3, 12, and 48 h. Red block is up-regulated DEGs. Green block is down-regulated DEGs. (**B**) PCA (principal component analysis) plot of the first two components for genes of ROS network. The dot color scheme represents different cotton accession (TM1, red; M85, yellow; CRI12, blue.). (**C**) Correlation coefficients of TM-1, CRI12, and M85. (**D**) RT-qPCR analysis of selected ROS genes, 1, 2, and 3 refers to the libraries, yellow-upregulated, blue: down-regulated, and black: no expression.

**Figure 8 ijms-20-01863-f008:**
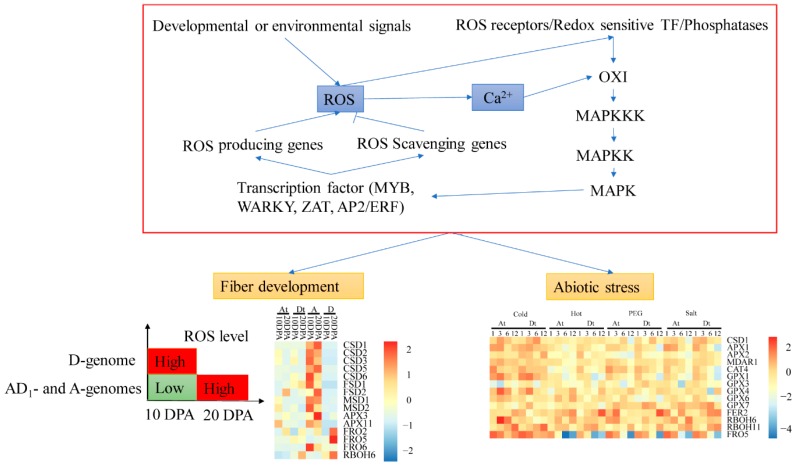
ROS network mediated signal pathway to illuminate the mechanism of fiber development and regulation of abiotic stress in *Gossypium*. Red box indicates modulation of ROS level by the reactive oxygen gene network of plants. ROS network regulated fiber development and abiotic stress by control ROS level, but it is not clear about how to do it. RNA-seq data of TM-1, CRI12 and M85 proved that serine/threonine kinase (OXI), and motigen-activated protein kinase (MAPK) cascades might involve in the alkali-salt stress (one of abiotic stresses) response which ROS network mediate. Under fiber development-box, heatmap of some ROS network genes indicted genes expression levels at 10 and 20 DPA (A, *G. arboreum*; D, *G. raimondii*; At, *G. hirsutum* A subgenome; Dt, *G. hirsutum* D subgenome) by R package (Pheatmap, scale = “row”). Under abiotic stress-box, heatmap indicated value of genes expression levels fold change (log_2_ (FC)) under cold, hot, dehydration and salt stress by Pheatmap package.

**Table 1 ijms-20-01863-t001:** The number of synteny blocks which including 1–20 gene number, respectively.

	Number of Synteny Block			
NGEB	At vs. AA	At vs. DD	Dt vs. AA	Dt vs. DD
1	136 *	63	140	58
2	19	10	19	13
3	3	11	4	8
4	2	7	2	3
5	1	7	1	4
6	0	3	0	3
7	0	1	0	2
8	0	2	0	2
9	0	0	0	2
10	0	1	0	0
11	0	0	0	2
15	0	0	0	0
16	0	0	0	1
20	0	0	0	1

NGEB: number of genes per synteny block; * for example, there are 136 synteny blocks which include one ROS gene between At subgenome and A genome.

**Table 2 ijms-20-01863-t002:** Phenotype of six *G. hirsutum* accessions at 48 h post to complex alkali-saline stress.

Accession	RL	SL	Area of Damaged Blades			
			First Leaf *	Second Leaf *	Third Leaf *	Fourth Leaf *
Latifolium 130	0.90 ^a^	0.80 ^c^	0.97 ^a^	0.73 ^a^	0.65 ^a^	0.00 ^a^
Latifolium 32	0.92 ^a^	0.97 ^a,b^	0.17 ^b^	0.03 ^b^	0.00 ^b^	0.00 ^a^
Latifolium 40	0.87 ^a^	0.89 ^b,c^	0.40 ^b^	0.20 ^b^	0.23 ^a,b^	0.03 ^a^
Marie-galante 85	0.92 ^a^	1.02 ^a^	0.32 ^b^	0.03 ^b^	0.00 ^b^	0.00 ^a^
CRI 12	0.98 ^a^	0.68 ^d^	0.38 ^b^	0.08 ^b^	0.02 ^b^	0.00 ^a^
CRI 16	1.12 ^a^	0.94 ^a,b^	0.38 ^b^	0.10 ^b^	0.05 ^b^	0.05 ^a^

Letters following the number represents significance at 0.05 probability. Different letters a, b, c and d in a column mean statistically significant difference at the 5% level. * Represents the value of leaves damaged rate (Area of damaged leaf/Area of leaf). First leaf means that the expansion of the first true leaf.

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
