# Peer review of "Deep Transcriptome Analysis Reveals Reactive Oxygen Species (ROS) Network Evolution, Response to Abiotic Stress, and Regulation of Fiber Development in Cotton"

_ijms, 2019, doi:10.3390/ijms20081863_

Round 1

Reviewer 1 Report

I recommend doing extensive revision and re-evaluation of the manuscript for the following reasons:

I have gone through the manuscript several times and found several Key issues:

-          A clear, new and full hypothesis is not presented and, without a robust hypothesis. In addition, hypotheses are important to define treatments, evaluate the methods used and guide the discussion section. 

-          The manuscript was poorly written particularly the part of results.

-          P 3, line 96, I think the words “these proteins” refer to the identified 1036 proteins so the previous sentence on the 279 PER proteins should be moved to the end of the paragraph.

-          Lots of explanations are inserted in the results section as in 2.3 and 2.4. These explanations should be moved to the discussion section.

-          I do not understand why authors did not measure the physiological parameter at the same intervals that taken for cDNA??

-          The supplementary tables are not arranged within the text as Table S2 is in result section 2.1 however Table S1 is mentioned in section 4.7

-          The resolution of Figures 2, 3 and 4 are quite low

-          There is a title for parameters in each graph as in Figure 2, I think this is very confusing and not useful because it is already mentioned on Y-axis definition

-          Correct the word “ortholog” on the Y-axis, Fig 2C to “orthologous”

-          Definition units on Y-axis are not complete as in Figure 5 “U/g or umol/g”. please indicate g FW or g DW and please use the micromole symbol instead of “u”

-          Legend of figures should be more descriptive so that the reader does not need to go back to the text to interpret it. Also, there are no red or blue arrows in Figure 5 as indicated in the legend, do the authors mean red and blue columns? And the time point “48h” is not indicated in the legend. The significance of the statistical analysis and the P value also is not indicated.

-          The cutoff parameters that used in defining the number of DEGs in Figure 6 are not indicated either in fig 6 legend or in the methodology. And I think doing Venne diagram for up- down-regulated genes will be very informative to indicate the common DEGs at each point of time in the examined species.

-          It was indicated in methods section 4.6 that the root and leaves samples collected at four time points, at 0h, 3h, 12h and 48h however the figures of gene expression showing other two timepoints 10 and 20 DPA??

-          I think it is mandatory to validate the RNA-Seq expression values at least for some examined genes by real-time PCR.

-          The abstract contains too many undefined terms as Ka however Ka/Ks ratio is not mentioned, and the purpose of used methodology has not been addressed

-          Section 4.2, line 447: the link for KaKs_Calculator package or the software involving the package was not involved.

-          Section 4.5, lines 470-480: names and details of the used kits in the physiological analysis and enzymes are not mentioned. The details as company name, cat. No., country of origin. “SPAD”, it is not “a photosynthesis indices” as authors mentioned but it is a unit measuring greening/chl of leaves.

-          The manuscript contains no conclusion.

-          The scientific names are not in italic in the references section.

-          Generally, I think that the results part needs to be re-write because there is a lot of redundancy in the way it's written and too many grammatical, spelling or punctuation errors. For example: genera not genus as in page 1, line 37, words or terms repeated thoroughly in the manuscripts as: transcriptional analysis not transcription analysis, sub-genome not sub genome, P 10, line 272: the sentence “It suggested that alkali-salt stress have not result lipid peroxidation …..”

-          I think the manuscript of Zhang et al. in Scientific Reports volume 8, Article number: 13527 (2018) is very important and it was not cited.

For these reasons, I do not recommend the manuscript, in the present form, for publication in the International Journal of Molecular Sciences.

Author Response

Reviewer 1.

1. I have gone through the manuscript several times and found several Key issues: A clear, new and full hypothesis is not presented and, without a robust hypothesis. In addition, hypotheses are important to define treatments, evaluate the methods used and guide the discussion section.

Response: this research aimed at revealing the ROS network evolution, response to abiotic stress and regulation of fiber development in cotton, based on genome and transcriptome-wide analysis.

2. The manuscript was poorly written particularly the part of results.

Response: The entire result section has been reorganized and citation done effectively.

3. P 3, line 96, I think the words “these proteins” refer to the identified 1036 proteins so the previous sentence on the 279 PER proteins should be moved to the end of the paragraph.

Response: I have already modified this sentence.

4. Lots of explanations are inserted in the results section as in 2.3 and 2.4. These explanations should be moved to the discussion section.

Response: adjustments made and a number of sections moved to the discussion part.

5. I do not understand why authors did not measure the physiological parameter at the same intervals that taken for cDNA??

Response: the physiological parameters we evaluated in order to understand the differences between the various genotypes used, being morphologically they are more less the same. Thus, in order to get the differences, longer stress exposure was necessary.

6. The supplementary tables are not arranged within the text as Table S2 is in result section 2.1 however Table S1 is mentioned in section 4.7

Response: Sorry for this, a mistake on our part, all figures, tables and supplementary files have been rearranged and orderly well as per the manuscript.

7. The resolution of Figures 2, 3 and 4 are quite low

Response: The resolutions improved and tiff formats provided along side the manuscript

8 There is a title for parameters in each graph as in Figure 2, I think this is very confusing and not useful because it is already mentioned on Y-axis definition

Response:correction done

9. Correct the word “ortholog” on the Y-axis, Fig 2C to “orthologous” Response: Changed 10. Definition units on Y-axis are not complete as in Figure 5 “U/g or umol/g”. please indicate g FW or g DW and please use the micromole symbol instead of “u”

Response: corrected

10. Legend of figures should be more descriptive so that the reader does not need to go back to the text to interpret it. Also, there are no red or blue arrows in Figure 5 as indicated in the legend, do the authors mean red and blue columns? And the time point “48h” is not indicated in the legend. The significance of the statistical analysis and the P value also is not indicated.

Response: adjustments done

11. The cutoff parameters that used in defining the number of DEGs in Figure 6 are not indicated either in fig 6 legend or in the methodology. And I think doing Venne diagram for up- down-regulated genes will be very informative to indicate the common DEGs at each point of time in the examined species.

Response: the parameters of DEGs identification was described in the methodology (section 4.7) and section 2.7. EBSeq software was used to identify the differential expression genes by Fold change ≥ 2 and FDR ≤ 0.01 (False Discovery Rate). FDR was corrected using Benjamini-Hochberg method by p-value.

12. It was indicated in methods section 4.6 that the root and leaves samples collected at four time points, at 0h, 3h, 12h and 48h however the figures of gene expression showing other two timepoints 10 and 20 DPA??

Response: adjustements done being both stress and fiber development was factored in this research work.

13. I think it is mandatory to validate the RNA-Seq expression values at least for some examined genes by real-time PCR.

Response: Done and the RNA seq. results and the RT-qPCR were in agreement.

14.The abstract contains too many undefined terms as Ka however Ka/Ks ratio is not mentioned, and the purpose of used methodology has not been addressed

Response: nonsynonymous rate (Ka) and synonymous rate (Ks) were frequently-used terms in evolution analysis, to estimate divergence time of different species or genes. And, ka/ks could be used for Selective pressure analysis. I think it is wide known, so I few describe in the abstract.

15. Section 4.2, line 447: the link for KaKs_Calculator package or the software involving the package was not involved.

Response: the link.added

16. Section 4.5, lines 470-480: names and details of the used kits in the physiological analysis and enzymes are not mentioned. The details as company name, cat. No., country of origin. “SPAD”, it is not “a photosynthesis indices” as authors mentioned but it is a unit measuring greening/chl of leaves.

Response: correction done

17. The manuscript contains no conclusion.

Response: Added

18. The scientific names are not in italic in the references section.

Response: all corrected

19. Generally, I think that the results part needs to be re-write because there is a lot of redundancy in the way it's written and too many grammatical, spelling or punctuation errors. For example: genera not genus as in page 1, line 37, words or terms repeated thoroughly in the manuscripts as: transcriptional analysis not transcription analysis, sub-genome not sub genome, P 10, line 272: the sentence “It suggested that alkali-salt stress have not result lipid peroxidation …..”

Response: We appreciate, but we have carried out extensive review and we hope the manuscript in its current form is acceptable.

Reviewer 2 Report

The authors describe a research focusing on genes involved in ROS in cotton with analysis of both public and self-generated data. The manuscript displays results from multiple aspects in evolution, syntenic analysis and RNA-seq data analysis. The authors try to use those to get an understanding on ROS gene network in cotton. There were lots of RNA-seq results have been published in cotton, and authors extracted those ROS genes to serve on their research topics. However, to make the manuscript to be publishable, there are still lots of stuff authors should make updates on. Language written in the manuscript needs to be polished a lot, and it is always better to search other people’s publications online to confirm the correct using way, like “subgenome”. The authors analyzed lots of data either from published results or their own generated data, to make the story as complete, they should describe them clearly and avoid any places not informative. The authors seem to be confused on “discussion” and “results” session, and should display words in appropriate places. Also, they should be strict on writing methods session, any confusion will lead to very lethal misunderstanding. Authors are too urgent to make a conclusion, for somewhere I think there is no enough evidence to support that, which should be moved to discussion session. Similar errors are easily to be captured in the manuscript and authors should make big updates on that. But authors have a relatively clear thought on analyzing those data, they need to display those to be clear. Below are some points I suggest authors to update in their next version and hope the manuscript will look much better.

1. please correct all places referring “sub genome” to “subgenome”

2. authors should specify either citations or link for “KaKs_calcultor” in the methods section and which model determined to choose for estimating KaKs.

3. line 90-91, needs to specify what is the evolution theory of the tetraploid cotton and how this proportion can support this theory.

4. line 125 “…this indicates their integral role they play within the plants”

5. line 189-200, too redundant and not necessary for results session and should move to discussion. In this same session, there are too many conclusions just based on extension from other publications, should move all to discussion, and restructure this paragraph.

6. line 224-225, I didn’t see any GWAS result analysis in this paper, if authors did not do GWAS or RNA-seq as they mentioned in this sentence, please add proper citations to support your conclusions.

7. please add y-axis label to figure 3, although you mentioned it in the legend. And the scale in y axis is so different, it will be very misleading to authors, I don’t think this current version of figure is meaningful.

8. please increase the size of dots in figure 6B, and add figure legend to the blank area of the figure. There is no any patterns that I can see from this PCA plot currently, and it is better to zoom in the figure. Since Figure 6 just shows a data validation, there is no need to use them as main figures and should be moved to supplementary information.

9. two figure 1? please be very carefully checking this type of errors before submission. Also, please rewriting everything in “figure 7”, now it looks messy.  

10. in table 2, authors mention “letters following the number represents significance at 0.05 probability”, but did not mention what a,b and c means, that would be critical for readers.

11. how many replicates and which accession did authors use for performing RNA-seq analysis? If just one, and there is no any further experimental validation, I am highly doubt the repeatability of the results.  Must describing clearly how to conduct the salt-alkali stress treatment.

12. for figure 5, since majority of subpanels shows the difference in between control and salt condition, adding the subpanel A in this figure is very misleading, authors should think about how to present their data in this scenario. And please only showing the traits with significant difference, for the ones do not have this pattern, just move to supplementary information.

13. In the methods, authors mentioned that they used Illumina Hiseq 2500, but in the result, they mentioned using Illumina Hiseq 2000. Manuscript should not just try to display something sounds like scientific but no any self-inspection process on organizing language. This type of mistake is very bad. And authors also not mention if using single-end? paired-end? how many base pair per raw read? which stage of samples did they use for RNA extraction and sequencing?

14. In figure 4, there are legends mentioning subpanel A and B, but there is actually no any A and B labels in figure 4, I don’t know what they refer to in the figure. Also, for any genes in those conditions, there must have different variations in those conditions and authors should demonstrate how confident is their conclusions, like drawing similar bar plots for some housekeeping genes. To me, there is no any special patterns for those genes.

15. the abstract of the manuscript should be improved to balance their results, now it sounds like emphasize too much on Ka/Ks results, but very few on talking about others.

16. In figure 1, should not mixed up figure and table as one big figure, if authors still want to use the table, make it as separate. 

Author Response

Reviewer 2.  

1.      please correct all places referring “sub genome” to “subgenome”

Response: corrected

2.      authors should specify either citations or link for “KaKs_calcultor” in the methods section and which model determined to choose for estimating KaKs.

Response: The link added

3.      line 90-91, needs to specify what is the evolution theory of the tetraploid cotton and how this proportion can support this theory.

Response: correction done

4.      line 125 “…this indicates their integral role they play within the plants”

Response: the sentence adjusted

5.      line 189-200, too redundant and not necessary for results session and should move to discussion. In this same session, there are too many conclusions just based on extension from other publications, should move all to discussion, and restructure this paragraph.

Response: correction done

6.      line 224-225, I didn’t see any GWAS result analysis in this paper, if authors did not do GWAS or RNA-seq as they mentioned in this sentence, please add proper citations to support your conclusions.

Response: Added

7. please add y-axis label to figure 3, although you mentioned it in the legend. And the scale very misleading to authors, I don’t think this current version of figure is meaningful.

Response: correction done

8. please increase the size of dots in figure 6B, and add figure legend to the blank area of the figure. There is no any patterns that I can see from this PCA plot currently, and it is better to zoom in the figure. Since Figure 6 just shows a data validation, there is no need to use them as main figures and should be moved to supplementary information.

Response: correction done

9. two figure 1? please be very carefully checking this type of errors before submission. Also, please rewriting everything in “figure 7”, now it looks messy.

  Response: Thanks, correction done

10. in table 2, authors mention “letters following the number represents significance at 0.05 probability”, but did not mention what a,b and c means, that would be critical for readers.

Response: adjustment done

11. how many replicates and which accession did authors use for performing RNA-seq analysis? If just one, and there is no any further experimental validation, I am highly doubt the repeatability of the results.  Must describing clearly how to conduct the salt-alkali stress treatment.

Response: I have contributed three cDNA library of every sample. CRI12 and M85 were used for performing RNA-seq. CRI12, M85 and TM-1 (public data) RNA-seq data were used for analysis. Those data proved the reliable data and the conservative expression model of ROS genes in G. hirsutum.

12. for figure 5, since majority of subpanels shows the difference in between control and salt condition, adding the subpanel A in this figure is very misleading, authors should think about how to present their data in this scenario. And please only showing the traits with significant difference, for the ones do not have this pattern, just move to supplementary information.

Response: thanks but we feel the results are integral and showld within the text

13. In the methods, authors mentioned that they used Illumina Hiseq 2500, but in the result, they mentioned using Illumina Hiseq 2000. Manuscript should not just try to display something sounds like scientific but no any self-inspection process on organizing language. This type of mistake is very bad. And authors also not mention if using single-end? paired-end? how many base pair per raw read? which stage of samples did they use for RNA extraction and sequencing?

Response: In this research, Illumina Hiseq 2500 was used to obtain the RNA-seq data. Correction done

14. In figure 4, there are legends mentioning subpanel A and B, but there is actually no any A and B labels in figure 4, I don’t know what they refer to in the figure. Also, for any genes in those conditions, there must have different variations in those conditions and authors should demonstrate how confident is their conclusions, like drawing similar bar plots for some housekeeping genes. To me, there is no any special patterns for those genes.

Response: correction done

15. the abstract of the manuscript should be improved to balance their results, now it sounds like emphasize too much on Ka/Ks results, but very few on talking about others.

Response: Thanks adjustment done

16. In figure 1, should not mixed up figure and table as one big figure, if authors still want to use the table, make it as separate.

Response: all figures have been rearranged

Round 2

Reviewer 2 Report

The authors did a nice job on solving my questions, but I still have a few concerns

1> Fonts in the figure 1 are too small to see

2> Check the capitalization error in line 498

3> Sentence from 274 to 276, I check reference 37, but I did not see anything in reference 37 talked about GWAS has identified those genes. I highly recommend authors to double check all other references in the manuscript to make sure they are correct. 

Author Response

Open Review

(x) English language and style are fine/minor spell check required 

Response: Thanks so much, we have revised all typos and corrected all the grammatical errors.

Yes

Can be improved

Must be improved

Not applicable

Does the introduction provide sufficient background and include   all relevant references?

(x)

( )

( )

( )

Response: Thanks

Is the research design appropriate?

(x)

( )

( )

( )

Response: Thanks

Are the methods adequately described?

( )

(x)

( )

( )

Response: We have elaborated on salt-alkali treatment method   to make it much easier for readers and future replication

Are the results clearly presented?

( )

(x)

( )

( )

Response: Figures improved, more so Figure 1 which had very   small font size, making it difficult for readers, oth PDF and TIFF formats   are provided

Are the conclusions supported by the results?

( )

(x)

( )

( )

Response: We appreciate the concern, but the indepth   information provided in the summary provide overall finding and our   suggestions for more research on the ROS network genes

Comments and Suggestions for Authors

The authors did a nice job on solving my questions, but I still have a few concerns

1> Fonts in the figure 1 are too small to see

Response: Changed

2> Check the capitalization error in line 498

Response: Corrected

3> Sentence from 274 to 276, I check reference 37, but I did not see anything in reference 37 talked about GWAS has identified those genes. I highly recommend authors to double check all other references in the manuscript to make sure they are correct. 

Response: Thanks, all citations are appropriate and further information added to qualify our statement in this section.